# Which Child with Asthma is a Candidate for Biological Therapies?

**DOI:** 10.3390/jcm9041237

**Published:** 2020-04-24

**Authors:** Andrew Bush

**Affiliations:** Imperial College & Royal Brompton Harefield NHS Foundation Trust, London SW£ dNP, UK; a.bush@imperial.ac.uk; Tel.: +44-207-351-8232

**Keywords:** airway eosinophilia, blood eosinophil count, omalizumab, mepolizumab, exhaled nitric oxide, induced sputum, allergic sensitization

## Abstract

In asthmatic adults, monoclonals directed against Type 2 airway inflammation have led to major improvements in quality of life, reductions in asthma attacks and less need for oral corticosteroids. The paediatric evidence base has lagged behind. All monoclonals currently available for children are anti-eosinophilic, directed against the T helper (TH2) pathway. However, in children and in low and middle income settings, eosinophils may have important beneficial immunological actions. Furthermore, there is evidence that paediatric severe asthma may not be TH2 driven, phenotypes may be less stable than in adults, and adult biomarkers may be less useful. Children being evaluated for biologicals should undergo a protocolised assessment, because most paediatric asthma can be controlled with low dose inhaled corticosteroid if taken properly and regularly. For those with severe therapy resistant asthma, and refractory asthma which cannot be addressed, the two options if they have TH2 inflammation are omalizumab and mepolizumab. There is good evidence of efficacy for omalizumab, particularly in those with multiple asthma attacks, but only paediatric safety, not efficacy, data for mepolizumab. There is an urgent need for efficacy data in children, as well as data on biomarkers to guide therapy, if the right children are to be treated with these powerful new therapies.

## 1. Introduction

The purpose of this review is to give a clinically focused update on the approach to the child with asthma for whom the prescription of a biologic is being considered (omalizumab or mepolizumab, the only ones currently licensed in children), in order to appropriately select those children who need these expensive and invasive medications, to highlight the important differences between adult and paediatric severe asthma with regard to the use of biologicals and to summarise the paediatric biologic data currently published.

The *Lancet* asthma commission has highlighted that the word “asthma” is an umbrella term comprising numerous endotypes [1]. Personalised asthma medicine was first practiced by the late Dr Harry Morrow-Brown, who used his medical school microscope to show that only those patients with sputum eosinophilia responded to prednisolone and inhaled beclomethasone. This meant that two of the most effective asthma therapies that we have were not lost. This valuable lesson, a really early attempt at personalised medicine, was lost to the asthma community in the excitement at the efficacy of inhaled corticosteroids (ICS), which were widely and often indiscriminately prescribed. When the anti-interleukin(IL)-5 monoclonal mepolizumab became available, it was again prescribed indiscriminately in adult asthma and was initially thought to be ineffective [2,3]. Fortunately, the obvious fact that anti-T-helper 2 (TH2) strategies would likely not work in non-eosinophilic asthma was appreciated, and the benefits of mepolizumab in attack prone, eosinophilic adult asthmatics was appreciated [4,5].

So as an example, the absolutely critical importance of personalised therapy has not been lost on the cystic fibrosis (CF) community. The knowledge of the different classes of CF genes [6] led to the discovery of Ivacaftor, which was dramatically effective (improved weight, lung function and quality of life, sweat chloride concentration halved) in Class III gating mutations [7]. Had Ivacaftor been given to all patients with a wet productive cough, or even all patients with CF, it would have been discarded as inactive. There is an obvious lesson here for the asthma community—unless and until we really understand pathways to disease, we are at risk of discarding important therapies.

The data and indications for mepolizumab and other biologicals has been summarised recently by the ERS/ATS Task Force, but these are largely in adults [8], for whom there have been major benefits in terms of better quality of life, fewer asthma attacks, and less requirement for oral corticosteroids. Whether the patient is eligible for an anti-Type 2 inflammation biologic is usually determined by the peripheral blood eosinophil count, which in adult studies at least, has been shown to be a good surrogate for airway eosinophilic inflammation [9]. However, even in adult studies, the correlation between a TH2 high signature in bronchial epithelial cells and elevation in blood eosinophils and exhaled nitric oxide (F_E_NO) is not good [10] and periostin, now being discarded even in adult medicine, cannot be used in children because it is secreted by growing bone. So in summary, anti-TH2 strategies are deployed in adult medicine if there is an elevated blood eosinophil count, on the assumption that airway phenotypes are stable. The tacit assumption is that eosinophilia equates to TH2 pathway activation; but even in adults, non-TH2 eosinophilic phenotypes are well described in U-BIOPRED, related to genes encoding metabolic pathways, ubiquitination and mitochondrial function [11]. We discuss these and other assumptions in more detail below.

Currently, only two biologicals (omalizumab, mepolizumab) are licensed in children age six years and over for severe asthma. There are extensive paediatric omalizumab data, but for mepolizumab, extrapolation from adult studies comprise the bulk of our information; and extrapolation from adults to children is dangerous. In this review, we explore the following issues, which are highly relevant to the role of biologicals in children:Is the eosinophil always the “bad guy” or could there be a down side to the aggressive, anti-eosinophil strategies which have been effective in adults?Is paediatric severe, therapy resistant asthma (STRA) the same as adult disease?What is a truly severe disease in childhood, in other words, is it only children with STRA who should receive these medications?How should we evaluate children referred for biological therapies?What are the paediatric data on the biologicals, and how do we match the right biological to the right child?

The definition of STRA combines the pharmacological criteria in Table 1 together with a failure to identify any reversible factors or co-morbidities on detailed assessment (below), in other words, uncontrolled asthma even despite all basic management being optimised. We conclude with suggestions as to how the present unsatisfactory, often non-evidence-based situation can be rectified.

## 2. The Eosinophil: A Janus Cell, Facing Both Ways?

The eosinophil has long been considered the effector cell in Type 2 inflammation driven asthma, but potential important beneficial roles are often not considered, and there may be developmentally important roles. Immunological effects include B-cell priming and maintenance of memory plasma cells [12,13], and antigen-presenting functions in the intestine [14]. Adipose tissue eosinophils participate in beige fat thermogenesis and glucose homeostasis through regulation of alternatively activated macrophages [15,16]. At least in murine models, there is evidence that eosinophils possess significant antiviral effects, and enhancing the eosinophilic response inhibits experimental influenza and respiratory syncytial viral infection [17].

The eosinophil is important in immunity to parasites, and this may be particularly important in low and middle income (LMIC) settings. In this context, it should be noted that the predictive power of blood eosinophil counts for monoclonal responses may be less good than in a high income setting, although this has yet to be tested due to poor availability of these medications in LMICs.

In summary, the potential beneficial roles of the eosinophil should be considered in the developmental and geographical context of the individual patient when assessing the risks and benefits of anti-IL5 therapy.

## 3. Adult and Paediatric STRA: Similarities and Differences

The question arises as to whether Type 2 inflammation is important in paediatric STRA. Our large series of carefully characterised children with STRA who underwent bronchoscopy showed that most, but by no means all, had airway mucosal and bronchoalveolar lavage (BAL) eosinophilia [18]. To our surprise, evidence of TH2 activation was scant. Induced sputum supernatant was positive for IL5 in only 8/41 patients; BAL was interrogated using both Luminex and Cytokine Bead Array platforms, and in the fifty samples available, ten were positive for IL4, and eight for IL5 and IL13. Immunohistochemistry demonstrated more IL5 positive mucosal cells in controls, and equal numbers of IL13 positive cells in the two groups. It was difficult to conclude that TH2 inflammation was of major importance in this group; possibly at an early stage of the disease TH2 inflammation had played a role, but the pathway is steroid sensitive and all these patients were being prescribed high-dose ICS. Our subsequent studies have focussed on the possible role of the epithelial alarmin IL33 as a steroid-resistant cytokine implicated in the pathology of STRA [19,20]. These findings are in accord with the other studies [21,22].

The USA Severe Asthma Research Program studied (BAL) supernatant and alveolar macrophages in 53 asthmatic children, of whom 31 were thought to have STRA, and 30 non-smoking adults [21]. They analysed a total of 23 cytokines and found no differences between the groups for any individual cytokine, but by using linear discriminant analysis, five cytokines were able to differentiate between mild asthma, severe asthma and healthy controls: these were growth-related oncogene (GRO), RANTES (CCL5, regulated upon activation, normal T cell expressed and presumably secreted), IL12, Interferon (IFN)-γ and IL10. They also concluded that there was no TH2 signal (nor indeed classical signature TH1 cytokines) in severe paediatric asthma.

These observations have recently been taken further in a recent manuscript from the USA [22]. This group analysed bronchoalveolar lavage (BAL) (*n* = 68) samples from 52 children age 0.5–17 years with STRA, not all of whom were allergic. They found that memory CCR51 TH1 cells were enriched in BAL, and many viruses and bacteria were detected. Furthermore, TH17-associated mediators (IL23, MIP 3a/CCL20) were highly expressed but TH2 cells were not prominent. TH2 cytokines were detected, and correlated with total IgE and IL5 correlated with BAL eosinophil count. IL5, IL33 and IL28A/IFNl2 were increased only in multi-sensitized children. Overall, there was a dominant TH1, not TH2 signature, with multiple bacteria and viruses being present, irrespective of allergic status.

Overall, there is considerable evidence suggesting that STRA that has been treated with high-dose ICS may, in many cases, not be a TH2-driven disease. Of course, the most important question is not whether anti-IL5 strategies should work but whether they do work. However, these data underscore the need for trials in children, and that it is not acceptable to extrapolate from adult studies.

## 4. Are Sputum Phenotypes Stable in Paediatric Asthma?

The supposition has been that sputum phenotypes are consistent over time in adults. In the only paper in children studying sputum phenotypes longitudinally [23], sputum phenotype changes unrelated to change in prescribed treatment were very common; 20/42 (48%) children with severe asthma exhibited more than one sputum phenotype (eosinophilic, neutrophilic, mixed, pauci-inflammatory) over a one-year period; for mild–moderate asthma, 4/17 (24%) had different phenotypes on paired sputum samples. On consideration, perhaps this is unsurprising; a sputum phenotype does not exist in isolation, but in an environment. So, for example, a child with TH2-driven asthma may be pauci-inflammatory if taking ICS regularly, become eosinophilic if adherence tails off or the child is exposed to a large allergen load (as has been seen with thunderstorm asthma [24]) and neutrophilic if the child develops a viral lower respiratory tract infection. This underscores the need to go from phenotypes to endotypes, understanding the underlying pathophysiology and directing treatment accordingly.

Are adult biomarkers relevant to children? There have been far fewer biomarker studies in children compared with adults. A Cochrane review suggested that titrating treatment to levels of F_E_NO reduces the burden of asthma attacks [25], and a raised F_E_NO is predictive of a response to omalizumab in adults [26]. Peripheral blood eosinophil count is used in adult studies as a marker of airway eosinophilia when considering anti-TH2 monoclonal antibody therapy. The INFANT study in pre-school children [27], albeit in post-hoc analyses, showed that peripheral blood eosinophil count combined with evidence of aeroallergen sensitisation, predicted response to ICS. However, in our hands, there is a poor correlation between blood and sputum eosinophil count; 76/88 (86%) of our STRA patients had a normal blood eosinophil count, of whom 64 (84%) had airway eosinophilia [28]. The most recent U-BIOPRED data have demonstrated that biomarkers for transcriptomically measured airway Type 2 inflammation are insufficiently sensitive and specific even in adults [10].

Omalizumab response in adults is also predicted by a raised blood eosinophil count [26]. In our hands, a fall in F_E_NO in response to intramuscular triamcinolone was the best predictor of omalizumab response, in a small study which has yet to be replicated [29].

It should also be noted that F_E_NO and sputum eosinophil count are not interchangeable [30]. In one longitudinal study, 79 children (51 severe, 28 mild-moderate asthma) contributed 197 paired sputum and F_E_NO measurements. Upper limits of normal were defined as F_E_NO ≥ 20ppb and sputum eosinophils ≥2.5. In the cross-sectional study, 75% pairs were concordant; longitudinally, only 53% were consistently concordant, and 7% were consistently discordant. The relationship varied over time, with some children sometimes having a high F_E_NO and normal sputum eosinophils, and then the reverse pattern on a subsequent visit. Adult studies have suggested that a raised sputum eosinophil count reflects IL5 activity, and a raised F_E_NO that of IL13 [31], but this has not been tested as a selection criterion between different monoclonal strategies, or validated in children.

In summary, it is wrong to extrapolate endotypes from adults to children; and it is wrong to assume that biomarkers which may be valuable in adults have the same value in children. There is an imperative for us to do paediatric studies.

## 5. What Is True STRA in Children?

It cannot be over-emphasised that most children who are referred with “severe asthma” to tertiary centres just need to get the basics right [32]. The basic management must be assessed before considering whether the child has true STRA. If a child is not responding to low-dose ICS, the correct action is not to increase the doses but to ask “Why is this child not responding to what should work?” The answer is usually not because of unusual airway pathology, but due to failure to get the basics right. This is confirmed by thee key studies, and our own data. In the BADGER study [33], very few children got additional benefit from increasing their ICS dose above fluticasone 100 mcg/day; the greatest benefit was from adding inhaled long-acting β-2 agonists (salmeterol), and a few benefited from the addition of a leukotriene receptor antagonist (LTRA). A study determining whether azithromycin or montelukast was a better add-on treatment for children with uncontrolled asthma despite ICS and long-acting beta agonists ended in futility because most such patients were either not taking treatment or did not have asthma [34]. An inner city study to determine whether using F_E_NO to monitor asthma added value to standard guidelines was also futile because during the run-in period, when protocolised treatment was emphasised, the children improved so much that there was no real scope for further improvement [35]. Finally, we showed that at least half of all children referred to our severe asthma service just needed to get the basics right. So biological therapies are only exceptionally needed in children with asthma [32].

It is also clear that we need new concepts of severe asthma. Almost without exception, guidelines and consensus documents define severity by (usually arbitrary) levels of prescribed medication [36]. Table 1 shows the criteria used by the first ERS/ATS Task Force. The problem is that the level of prescribed medication relates only weakly to the domain of risk. Patients prescribed very low-dose medications may be at risk of a serious asthma attack, and even death, especially if they do not use the medications efficiently. This was clearly demonstrated by the UK National Review of Asthma Deaths [37], which showed that around half of asthma deaths were in those who would not have been classified as having severe asthma. The deaths were not related to difficult airway pathology, but to social and environmental factors, such as under-use of ICS [38], over-use of short-acting β-2 agonists [39], failure to attend routine asthma reviews, and frequent emergency care visits. The lesson from these data is that any definition of severe asthma based merely on levels of prescribed medication is not adequate. We also reported that most children referred for consideration of “beyond guidelines” therapy in fact only need to get the basics right [32]. Although many respond to guidance, some do not and remain at high risk. The important conclusion is that a definition of severe asthma solely based on levels of prescribed medication is inappropriate and environmental, and in particular social factors, must enter the definition even in adults; and this has particular relevance to deciding who should get biologicals (below).

## 6. How Should We Evaluate Children for Biological Therapies?

The protocols we use have been discussed in detail elsewhere [40,41,42], and are summarised in Figure 1.

The starting point is a child referred with respiratory symptoms that do not respond to asthma therapy. The first step is a detailed history and examination, with a focussed approach to testing, to exclude other diagnoses, such as vascular ring or bronchiectasis (Figure 2). The likeliest differential diagnoses will depend on geography—airway compression by tuberculous lymph nodes is rare in London, but common in high burden areas. If it is likely that the underlying diagnosis is indeed asthma, the next step is a multi-disciplinary team assessment (Table 2).

There is a focus on an assessment of social and environmental factors. The child is then placed into one or more of the overlapping categories asthma plus (co-morbidities, such as obesity, food allergy, exercise-induced laryngeal obstruction (EILO [43]), rhinosinusitis), difficult asthma (asthma which could be controlled if the basics can be got right; poor adherence, adverse environmental factors such as sensitization and exposure to allergens, passive or active smoking, and psychosocial factors) and STRA. On the basis of the findings, the child with STRA goes on to invasive airway phenotyping (below), and an individualised management plan is put in place for those with difficult asthma and asthma plus. The success of the plan is reviewed two months later; many children will have responded to their plan [44], and are able to reduce treatment with improved outcomes. However, there remain a group which we term refractory asthma plus (especially obesity with failed weight loss), and refractory difficult asthma (usually continued very poor adherence or unwillingness or inability to control adverse environmental factors, such as allergen exposure) [45]. It should be remembered that poor adherence may in fact reflect STRA, and that the child and family are not taking medications which have not worked. Children with refractory asthma will also undergo invasive airway phenotyping to determine treatment. Our previous thinking was that it was not justified to give an expensive biologic to children who are not using low-dose ICS efficiently or whose parents will not get rid of an allergenic pet; we now believe that it is wrong to penalise children for the lack of adequate parenting, and we need to keep them alive despite the parenting issues. We now conclude that refractory difficult asthma due to poor adherence is not a contraindication to the use of biologicals.

The invasive phenotyping protocol is summarised in Table 3. The aim is to answer the following questions:Is there ongoing airway inflammation, and if so, what is the phenotype/endotype?Is any inflammation present steroid sensitive? (For example, corticosteroids are very effective against eosinophilic inflammation, but not in neutrophilic disease)Is there a disconnect between the degree of inflammation and the level of symptoms?Is there evidence of persistent airflow limitation?

This is particularly relevant to the obese child with asthma, who may have TH2-driven airway inflammation [46], but this is not invariable [47], and may have symptoms related to dysanaptic airway growth, defined as a normal first second forced expired volume (FEV_1_), a raised forced vital capacity (FVC) and a low FEV_1_/FVC ratio [48] or secondary to systemic inflammation driven by IL6 [49]. It is obviously also essential to ensure that symptoms like breathlessness are in fact not due to deconditioning [50].

If airway phenotyping has shown eosinophilic inflammation (and ideally, a transcriptomic signature for TH2 inflammation) then treatment with either omalizumab or mepolizumab may be indicated. If the phenotype is pauci-inflammatory or neutrophilic, the options are much more limited. Clearly, if the endotype is not TH2, then anti-TH2 strategies are unlikely to be successful. Options might include Tiotropium [51] or a macrolide [8], with reduction of the ICS dose [52]. However, as with adults [10], we do not have the perfect biomarker for TH2 asthma, and there may be overlapping syndromes, for example, children with neutrophilic or pauci-inflammatory disease and one or more aeroallergen sensitivities. Children with asthma may show discordance between the TH2 markers F_E_NO and eosinophils, and this relationship may change over time in the same child [30]. In reality, we need to know that (for example) an IL5 endotype is driving the child’s asthma before prescribing an anti-IL5 strategy, but this is not practical in current clinical practice.

It might be argued that this investigation protocol is unnecessary, and that all children with evidence of TH2-high asthma should receive a biological. However, we believe this is not a correctly focused view. Firstly, the evaluation may reveal why the asthma is severe, such as poor adherence, which at least some parents may be eager to rectify. Secondly, if we are to understand the true efficacy of biologics in children, we need to differentiate between children who have true STRA, and those for whom biologics are a fire-fighting measure to prevent the child dying. The airway endotypes in these two groups are likely to be dissimilar. Finally, given the differences between the utility of biomarkers for TH2 inflammation between children and adults, and the lack of specificity even in adults, a bronchoscopic evaluation to determine that Type 2 inflammation is actually present is fully justified to prevent the child being submitted to a long course of futile and time-consuming therapy.

## 7. What Are the Paediatric Data, and Who Should Get What Biologic?

There are currently five biologicals licensed in adults. These are omalizumab (binding to the high affinity IgE receptor), mepolizumab and reslizumab (both binding IL5), benralizumab (binds to IL5 receptor α subunit) and dupilumab (binds to IL4 receptor α subunit, thus blocking IL4 and IL13). Tezepilumab, which binds to TSLP, is in Phase 2B trials. Of these, only two (mepolizumab and omalizumab) are licensed for children with asthma; dupilumab is licensed for children with atopic dermatitis.

### 7.1. Omalizumab

Although there are fewer randomised controlled trials in children than in adults, the Cochrane review [53] was able to summarise a lot of data and give evidence-based guidelines. In summary, the suggested indications were a total IgE between 76 and 1500, evidence of aeroallergen sensitisation and multiple asthma attacks. It should be noted that around 30% of our STRA patients have a total IgE above the range of eligibility. Furthermore, adult data show that the response to omalizumab may be equally good in those meeting IgE criteria but without aeroallergen sensitization [26]. Asthma attacks are reduced by Omalizumab (odds ratio 0.55, 95% confidence intervals 0.42–0.60 in 10 studies recruiting 3261 patients). The absolute reduction in attacks was from 26% to 16%. Admissions to hospital were also decreased: odds ratio 0.16, 95% confidence intervals 0.06–0.42 in four studies recruiting 1824 patients. The absolute reduction was 3% to 0.5%. There was also a reduction in the softer end-point of short acting beta2 agonist (SABA) usage (odds ratio 0.16, 95% confidence intervals 0.06–0.42, in four studies recruiting 1824 participants). The reduction in actual use was a mean of −0.39 puffs per day, 95% confidence intervals −0.55 to −0.24 in nine studies recruiting 3524 patients.

The ERS/ATS Task Force [8] has recently suggested that a blood eosinophils ≥260/μλ and F_E_NO ≥19.5 ppb are cut-offs to predict response to omalizumab in adults with severe allergic asthma, but both were conditional recommendations based on low-quality evidence (a single study [26]). There are no biomarker data sufficient to inform recommendations in younger children. The UK authorities have insisted on the need for a history of at least four attacks of asthma requiring oral corticosteroids before omalizumab can be prescribed. Given the overwhelming evidence that the best predictor of a future asthma attack and asthma deaths is a previous attack [37,54], this recommendation, which is based on “cost-effectiveness”, cannot be right for children. The other restriction is that compliance to standard medications is assured. This too is wrong; firstly, because although non-adherence can be confirmed, e.g., failing to pick up or cash prescriptions or failure of activation of an electronic monitoring device [55], adherence can only be assured by directly observed therapy, which is (a) not easy to set up in clinical practice, and (b) fraught with pitfalls. Secondly, as above, children should not be allowed to die because their parents will not ensure they are taking basic treatment.

Recent data have highlighted that omalizumab may also have significant anti-viral effects. Every autumn, when the new school year begins, there is peak in asthma attacks related to respiratory viruses, and likely also worse adherence during the school summer holidays to inhaled corticosteroids. This peak of admissions was abolished in a study of more than 400 inner city children [56]. This was taken further in a study of 478 children [57]. There was a four to nine month run-in, before they were randomised before autumn 2012 and 2013 to either four months of omalizumab, or a pre-autumn boost in the prescribed dose of inhaled corticosteroids, or placebo. The biggest beneficial effect as seen in children at Step 5, and the two active treatment strategies (inhaled corticosteroid boost, omalizumab) were equally efficacious in the overall study group. However, if the child had had an asthma attack during the run-in period, omalizumab was the best strategy. In in vitro work, they demonstrated that Omalizumab boosted peripheral blood mononuclear cell IFN-α responses to rhinovirus, and this was correlated with clinical response. The authors therefore recommended the use of omalizumab only in those children having asthma attacks despite step 5 therapy.

### 7.2. Mepolizumab

The ERS/ATS Task Force [8] has made recommendations for the use of the anti-IL5 monoclonal mepolizumab. They suggested that mepolizumab should be used in adults as add-on therapy for patients with severe uncontrolled asthma with an eosinophilic phenotype. This was a conditional recommendation based on, at best, moderate quality of evidence. Mepolizumab reduced asthma attacks and hospitalizations and led to reduction in dose in those prescribed maintenance oral corticosteroid. The effects on asthma control, quality of life and FEV_1_ did not achieve the minimal clinically important difference.

Drug-related adverse events were slightly higher in those assigned to mepolizumab. Disappointingly, although an entry criterion for the mepolizumab trials was an age of 12 years and older, there were only 34 participants who were not adults. In the 6–11 year age group, the data are largely safety and pharmacokinetic [18] with only minimal efficacy data. Enrolment for these studies was on standard adult criteria, namely blood eosinophils ≥300/μλ at screening or ≥150/μλ in the previous year, and the dose was 40mg if body weight <40kg, 100mg if >40kg. It was unclear how many children met the ERS/ATS definition of STRA [36]. Studied end-points were adverse events, blood eosinophil counts, annualised exacerbation rate and asthma control (ACQ/c-ACT). Less than 50 children have been studied in total, for a total of one year. The overall conclusions were that no new safety issues were raised, and there was a marked decline in peripheral blood eosinophil counts, similar to those seen in adults. Such conclusions on efficacy as could be drawn were promising, but further work is needed. However, despite the paucity of data in young children, mepolizumab has been approved in children age six and over by EMA (European Medicines Agency). A trial of mepolizumab is of course reasonable in children age six and over who meet the blood eosinophil criteria and are having attacks if they fail to respond to omalizumab or are not eligible because the IgE is too high, but the paediatrician should remain aware of the limitations of the efficacy data of this biologic.

Another potential role for mepolizumab, not studied at all in children, is as adjunctive therapy after an asthma attack. The UK National Review of Asthma Deaths (NRAD) has taught us that the month after an attack is the period of highest risk for asthma deaths [37]; and we know that uncontrolled Type 2 inflammation is also a risk factor. Hence, we need a trial to determine whether a single dose of mepolizumab before the child is discharged from hospital after a severe attack could improve outcomes.

## 8. Limitations of Current Clinical Trials

As highlighted above, there is a paucity of studies in children, and even those studies which recruited children over the age of 12, in practice recruited adults almost exclusively. This must not be allowed to continue. Trials have assumed that biomarkers which are important in adults are correct for children, but this may not be the case. They have assumed that the safety issues are the same in children and adults. We are doing our children a disservice by uncritically extrapolating from adults to children. We need an international collaboration to recruit large numbers of carefully characterised children, with a wide range of biomarkers prospectively measured so we can learn how to predict responders in the future. We also need studies comparing the biologics in children.

## 9. Summary and Conclusions: Where from Here?

It is clear that there are important differences between paediatric and adult STRA. Paediatric STRA is often eosinophilic, but frequently, there is no evidence that the TH2 pathway is in play. There are significant doubts that peripheral blood eosinophil count relate closely to eosinophilic airway inflammation; eosinophilic inflammation may not be TH2 driven, at least in many cases; sputum cellular phenotype may not be consistent over time, and the eosinophil may have developmental roles which mean that anti-eosinophil strategies may have unanticipated consequences in young children. It is therefore clear that extrapolating adult data into the paediatric age group is potentially hazardous. The inescapable conclusion is that we need trials in children, and the regulatory authorities need to hold Pharma to account to ensure these studies are done.

The prerequisites for such studies include the careful characterisation of the patients; are they asthmatics who could be controlled on low-dose ICS if they were used properly, and thus it is likely the TH2 endotype is relevant, or are they true STRA, in which case, multiple different endotypes are probably important? The first pressing need is to determine who should be prescribed omalizumab and who mepolizumab, as these are the two biologicals licensed for children. A high proportion of STRA children may be eligible for either, and we have no current means of determining the best course of action. At the moment, in such cases, most would start with omalizumab as the biological with which we have the most experience. A head-to-head comparison trial is urgently needed, and is shortly commencing in the UK [58]. However, there are an increasing number of biologicals relevant to the TH2 pathway which will also become licensed in children, and we urgently need biomarkers to enable us to choose between them on a rational basis, rather than doing a succession of N-of-1 trials, with all the issues of placebo effects. Furthermore, perhaps combinations may be better—for example, mepolizumab (anti-IL5) and dupilumab (anti-IL4/IL13) co-administered might be a better anti-TH2 strategy than either alone, but this must be tested. This also highlights that we need objective biomarkers of response; clearly, if a child with an attack-prone phenotype is started on biological and ceases to have attacks, the benefits are clear-cut, but some benefits are less objective.

Another real challenge to the paediatric respiratory community is the anti-Il13 (trakilizumab) story. This agent was (ethically correctly) studied in adults and was found to be ineffective, and discarded [59]. Given the differences outlined above between adult and paediatric STRA, is it conceivable that trakilizumab could be effective in children but not in adults? And how could we ethically do such studies in children? There may be two possible answers. The first is assessing the effects in developmentally appropriate, physiological models of asthma, such as the murine neonatal, house dust mite inhalational challenge model, which recapitulates the features of early onset allergic airways disease [60]. The second may be to define biomarkers for a subgroup of IL13 upregulated patients—conceivably those with high F_E_NO and low sputum eosinophil counts.

A further, as yet unsolved, challenge is how to halt the march from pre-school viral wheeze, with no evidence of TH2 inflammation, to atopic, school-age asthma [61]. We know from three definitive randomised controlled trials that an early use of inhaled corticosteroids is not the answer [62,63,64]. Could an early institution of therapy be directed more specifically at TH2 inflammation, such as mepolizumab? For this to be explored, we need better to understand the endotypes, and especially find biomarkers to predict those who are at risk of disease evolution; current predictive indices have a high negative predictive value, but their positive predictive value is much less useful [65,66].

In summary, the advent of the new biological agents have brought us to the edge of a new age in asthma management. If the benefits are to be realised, we must insist that clinical trials are performed in children, and specifically that regulatory authorities insist on the submission of a credible paediatric investigation plan as a condition of permitting adult studies to be done. We need to get cleverer, abandoning umbrella terms like asthma, and determining endotypes in a way that is practical in the clinic. In this way, we can match the child to the most appropriate treatment. However, the final word in all asthma manuscripts like this must be that most children do not need expensive novel therapies to control their asthma. If the basics are got right, and low-dose ICS are used regularly and correctly, then most asthma becomes a disease that is eminently treatable.

## Figures and Tables

**Figure 1 jcm-09-01237-f001:**
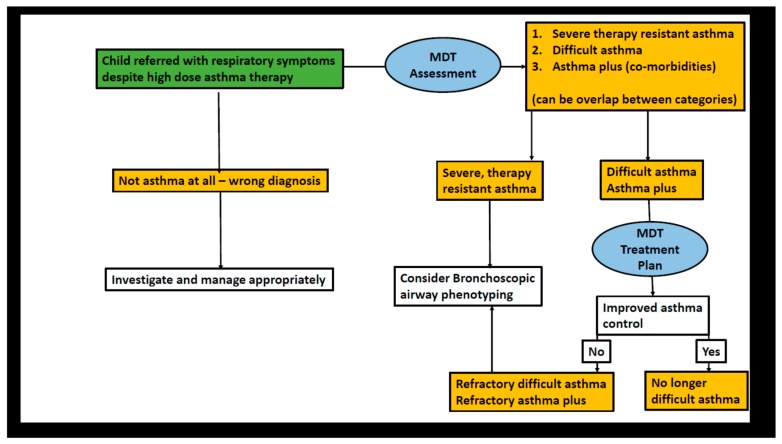
Flow chart for assessment of the child referred for assessment of asthma symptoms not responding to treatment. Abbreviation: MDT, multidisciplinary treatment.

**Figure 2 jcm-09-01237-f002:**
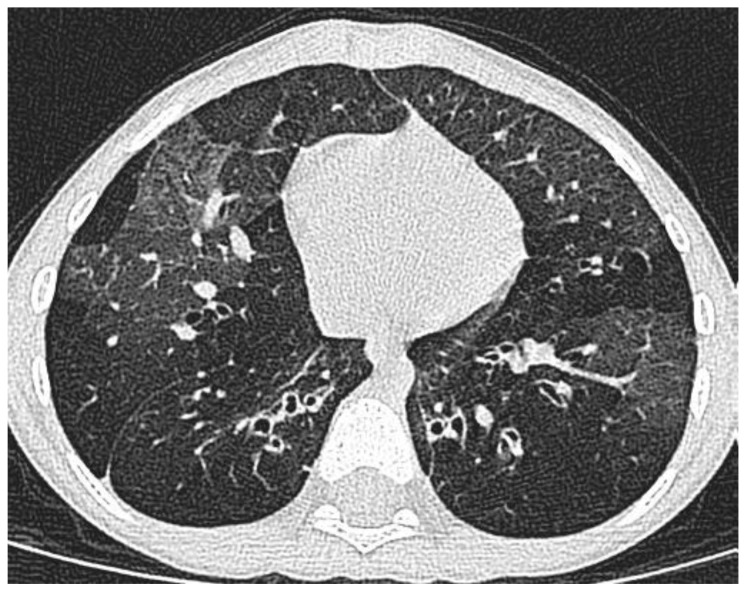
Not asthma at all. There is extensive large airway thickening and dilatation, with distal air trapping. This is bronchiectasis and obliterative bronchiolitis after a severe adenovirus infection.

**Table 1 jcm-09-01237-t001:** ERS/ATS Task Force definition of severe asthma [36]. The level of medication is combined with at least.

Level of Medication	Asthma Functional Deficit
Asthma which is only controlled or uncontrolled on therapy with ≥ 800 mcg/day BDP equivalent plus additional controllers (LABA, LTRA. Theophylline) or failed trials of these agents	Poor symptom control, e.g., Asthma Control Test (ACT) <20
	≥2 bursts of systemic corticosteroids (≥3 days each) in the previous year
	Serious exacerbations (≥1 hospitalisation or PICU stay) in the previous year
	Airflow limitation: FEV_1_ < 80% predicted following SABA and LABA withhold

Abbreviations: ACT, asthma control test; BDP, beclomethasone diproprionate; LABA, long acting beta2 agonist; LTRA, leukotriene receptor antagonist; PICU, paediatric intensive care unit; SABA, short acting beta2 agonist.

**Table 2 jcm-09-01237-t002:** Multidisciplinary assessment of severe, therapy-resistant asthma.

Issue to be Addressed	Tests Performed
Symptom pattern	ACT or c-ACT, asthma attacks and prednisolone bursts, unscheduled emergency visits; evidence of severity of symptoms at emergency presentation School attendance and impact of symptoms at school
Breathing pattern disorder	Physiotherapy assessment Consider asking parents to make a video of breathing pattern Consider laryngoscopy during exercise
Psychosocial factors	Questionnaires relating to treatment burden, anxiety and depression, quality of life
Physiology	Spirometry before and after bronchodilator Lung clearance index
Allergic sensitization	Total IgE skin prick tests and specific IgE to grass and tree pollen, house dust mite, cockroach, cat and dog, aspergillus, alternaria and cladosporium and any likely relevant other antigens Not food allergens unless a suggestive clinical history
Airway inflammation	FeNO Induced sputum cytospin for eosinophil count Peripheral blood eosinophil count
Nicotine exposure (tobacco or vaping, passive or active)	Urine cotinine
Medication adherence	Prescription uptake Serum prednisolone and theophylline levels if prescribed; serum inhaled corticosteroid levels if available (usually only in a research context) Electronic monitoring

Abbreviations: ACT, asthma control test; FeNO = Fractional expired nitric oxide; IgE = immunoglobulin E.

**Table 3 jcm-09-01237-t003:** Invasive airway phenotyping.

Tests	First Visit	Second Visit	Third Visit
**Non-invasive**	Assessment of current symptoms Spirometry before and after SABA LCI F_E_NO Induced sputum eosinophils	Assessment of current symptoms Spirometry before and after SABA LCI F_E_NO Induced sputum eosinophils	Assessment of current symptoms Spirometry before and after SABA LCI F_E_NO Induced sputum eosinophils
**Invasive**	Fibreoptic bronchocopy, BAL, endobronchial biopsy		
**Actions**	Intramuscular triamcinolone (steroid trial)	Assess steroid responsiveness Develop bespoke treatment plan	Assess response to treatment

Abbreviations: F_E_NO, fractional exhaled concentration of nitric oxide; LCI, lung clearance index; SABA, short-acting beta-2 agonist.

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
