# Peer review of "Which Child with Asthma is a Candidate for Biological Therapies?"

_jcm, 2020, doi:10.3390/jcm9041237_

Round 1

Reviewer 1 Report

The review by Dr Bush talks about TH2 therapies in pediatric asthma. Dr Bush nicely pointed out the fallacies in just looking into eosinophils as the target. In my opinion, the review is concise but can include a section on the issues highlighting the limitations of the current clinical trials ongoing or complete. This would benefit the readers of the paper and help in directing research. I would also recommend putting a summary table of the clinical trials ongoing or completed, the drug or treatment regime, the target and the primary and secondary expected or achieved outcomes. Please comment on including clinical trial (NCT02414854) and refer to phase II trials for Th17 subgroup of asthma. It would be good to put in a section at the end as future recommendations or future perspective rather than an overarching long conclusion.

Please check whether you have copyright permissions to republish tables and figures used in the manuscript. If you do you need to cite them accordingly. Other than that, the manuscript has several grammatical mistakes that prohibited my flow of reading and would need serious checking. Sentence structures in many places need to be corrected. I would also recommend summarising the reason for this review more concisely highlighting the current problem in the introduction. Lastly, I have pointed out the following errors in the whole manuscript which will be helpful for your revision.

  • line 13: Adult
  • line 16: asthma
  • line 19: ".[space] There", space is needed.
  • Line 26: world should be word.
  • Line 27: endotypes such as?
  • Line 34 asthma spelling.
  • Intro Para 1 needs significant rephrasing to give impact to readers.
  • Please establish link between asthma and CF at the start of para 2,. I feel lost going through it.
  • Line 48 eosinophil spelling.
  • Line 50, (.) after inflammation.
  • Again, in para 3 suddenly child asthma comes. The context of the review talks about childhood asthma and needs to be way ahead for readers to know what this review is going to be about.
  • I would like a paragraph summarised on the issues the review is meant for rather than a numbered list.
  • Line 77-79 doesn't make sense
  • Line 80- which urine models?
  • Summary statement should consider eosinophils and not just anti-IL5 therapy.
  • line 91, in my opinion, it is not a good way to start with a question mark.
  • Consider sentence formation in lines 95.
  • Please be consistent in writing your cytokines such as IL-5 or IL5.
  • Can move line 103 to the previous para.
  • What is BAL? I know its broncho-alveolar lavage but it would be ideal to note here that all such abbreviations be elaborate in their first appearance.
  • Please comment more on the TH1 status in line 117.
  • consider line 123 as per my previous comment.
  • line 124 needs restructuring.
  • line 135 same comment as before.
  • Line 137: FeNO.
  • move line 147-149 in the previous para.
  • section 1.3 first two sentences unclear.
  • line 168 name the studies.
  • line 169 what is the usual dose? please acknowledge it what is usual and what is increased.
  • line 173 and the study is written incompletely. please state the findings of this study.
  • please elaborate more on the treatment in lines 173-176.
  • line 193 asthma.
  • line 194 definition.
  • please check whether you need copyright permissions for use of table 1.
  • Please use larger font for figure 1.
  • Also I figure 1, try to use a decision based flow chart box (diamond).
  • line 209 there is no word as likeliest, please consider correcting.
  • line 210 diagnosis.
  • The entire second line and first line from line 207-211 doesn't connect. I don't see the link where tuberculosis is coming from.
  • Figure 2 is a CT I presume. Please reference here, if it is taken by you using the machine and controls. If by others, please refer to the journal where it is coming from (source).
  • For table 2, try to put in references if possible.
  • line 233 biologic is what you mean I guess.
  • Line 234 allergic I presume. Please re-wite the sentence to make it meaningful.
  • Table 3 and the paragraph following this should be with the first section after the introduction. In my opinion, it fits better there.
  • Please rephrase section 1.5 title without question.
  • line 365, I presume it to be anti-IL4 and anti-IL13. Please rectify.
  • line 387: that [space] clinical.

Reviewer 2 Report

The manuscript entitled “Which child with asthma is a candidate for biological therapies?” is an excellent manuscript that provides a fresh, engaging, and critical look about the selection of biologics in childhood asthma. It is well-written and discussed, and references are appropriate.

Nevertheless, some revision is needed.

  1. Line 9, the statement “All current monoclonals are anti-eosinophilic,” should be rephrased, as not all biologics are anti-eosinophilic, at least strictu sensu.
  2. Line 67, the author state: “Eligibility is usually determined by the peripheral blood eosinophil count”. As the paragraph is dealing with mepolizumab and other biologics, this statement is not clear to me. Could you be more specific, please?
  3. As the author is dealing with severe, therapy-resistant asthma, a brief definition is needed
  4. Line 243: The author asks: Is any present inflammation steroid-sensitive? It could be interesting to provide some clues about this topic
  5. Line 256: what about allergic children with pauci-inflammatory or neutrophilic inflammation? What about children with high FENO and low eosinophils? Can we consider these types of asthma as non-T2 asthma?
  6. Lines 281-283: The statement “adult data show the response to omalizumab is equally good in those meeting IgE criteria but without aeroallergen sensitization” is too categorical to me. I would suggest something like can be useful or similar, instead of equally good.
  7. Line 347: eosinophilic inflammation may not be TH2 driven, at least in many cases. This statement is not clear to me, particularly in asthma, please justify. Did you mean that it could be due to innate mechanisms, i.e. ILC2?
  8. It seems that the author is not in favor of using mepolizumab. Nevertheless, in a manuscript dealing with the selection of biologics, I miss some comments about possible uses of mepolizumab: what about asthmatic children not responding or with a partial response to omalizumab? What about the minority of nonallergic asthmatic children with eosinophilia?

Only some minor modifications are suggested:

  1. I would suggest using the term T2 asthma instead of Th2 asthma, thus englobing mechanism associated with innate immunity mediated by ILC2s
  2. Line 112: a closing parenthesis is lacking
  3. Line 174: Please, correct the typo using
  4. Line 212: Please, introduce a space in Table2
  5. Line 250: Please, correct the typo fiest
  6. Line 266: Please, correct the typo ustility
  7. Line 276: The term atopic dermatitis is more specific that eczema
  8. Line 365: Please, correct the typo anti-IL$
  9. Line 387: Please, introduce a space in thatclinical
  10. Please correct the typo tralikizumab
